# The Intersection of Spirituality, Religiosity, and Lifestyle Practices in Religious Communities to Successful Aging: A Review Article

**Deborah Tessitore McManus**

School of Nursing, Massachusetts College of Pharmacy and Health Sciences, Boston, MA 02115, USA; deborah.mcmanus@mcphs.edu

**Abstract:** Defining successful aging continues to be a challenge, given the more recent transition from a bioscientific definition to a more comprehensive and holistic perspective. The holistic perspective may include more subjective aspects of aging. Research has shown that certain factors, such as lifestyle practices of spirituality, religious practice, meditation, communal support, and purposeful living, may be as influential as genetic factors in helping aging adults diminish age-related limitations. Spirituality and religiosity as lifestyle practice resources may assist older adults to mitigate the circumstances of chronic disease and positively impact aging, life quality, and well-being. Religious and/or spiritual communities, such as Women Religious and other clergy and monks, may provide insight into specific practices that promote social exchanges, life meaning, meditative practice, daily prayer, belief in the divine, communal living, and homogeneity of lifestyle that ultimately promote successful aging and resiliency into older age. Research has shown that lifestyle factors may mitigate diseases such as Alzheimer's and other forms of dementia in older age. The population of older adults has grown consistently on a global level since the turn of the century. This article review seeks to examine aging and emphasize, through research, which lifestyle practices and communities may impact the experience of aging in a more beneficial manner.

**Keywords:** successful aging; aging; lifestyle practice; religiosity; spirituality; longevity; well-being; meditation; Catholic sisters; nuns; monks; older adults; religious communities

Advances in public health and medical technology allow someone aged 65 the expectation of living another two decades (Mahara et al. 2023). In this circumstance, aging research should provide directives and evidence that may support this notion. Despite this recent progress, other evidence suggests these last years may include disability given the rise in cancer and chronic diseases (Mariotto et al. 2020). Many older adults also require more services and support to maintain their quality of life in their older years. Directives that may assist the aging process outside of traditional medical technological advances are warranted. The ongoing exploration into the factors influencing successful aging and longevity has highlighted religious communities, particularly Catholic sisters, and nuns, also known as Women Religious, as exemplars of positive physical and psychological health outcomes compared to their lay counterparts. Research indicates that Catholic nuns experience extended lives and enhanced well-being, potentially attributed to lifestyle factors promoting successful aging. The multifaceted concept of successful aging, as defined by Rowe and Kahn (1987), involves longevity and the absence of disease and disability. Research suggests that Catholic nuns may exhibit successful aging as defined by Rowe and Kahn. Their positive health outcomes are often associated with cultural and lifestyle practices, including daily religious practice, prayer, communal living, robust social support, and spirituality. Their positive outlook on life, including embracing the end of life, contributes to overall well-being and a sense of community. The interrelated effects of religiosity and belief in the afterlife may also contribute to one's experience of aging.

The Nun Study, led by Dr. David Snowden, supports the notion that Catholic nuns lead happier and healthier lives. This longitudinal study, focusing on 678 Catholic nuns from the School Sisters of Notre Dame, reveals that nuns also live longer lives than the general population. Snowdon et al. (1999) examined linguistic abilities by analyzing journal entries made by nuns upon joining the convent. The study suggested that linguistic abilities might have served as a protective factor against dementia in older age within this religious community. The positive journal entries may also correlate to a longer lifespan, suggesting a potential link between a positive outlook and longevity. Another significant revelation from the study was the impact of lifestyle as a crucial determinant in mitigating Alzheimer's disease. Postmortem examinations of the nuns' brains did reveal pathological indications of Alzheimer's disease; for example, amyloid beta plaques and tau neurofibrillary tangles were present. The level of cognitive or physical impairment that was observed in the nuns throughout their lifetimes did not match the extent of neurodegeneration present in their brains. This suggests the existence of protective factors that safeguarded against cognitive decline despite the underlying pathology that was detected. The level of education reported by Catholic nuns, particularly those with a bachelor's degree or higher, was linked to lower rates of Alzheimer's disease and increased life expectancy (Snowdon et al. 1999).

The Religious Orders Study was another longitudinal, clinical–pathologic study of aging and Alzheimer's disease in older Catholic nuns, priests, and brothers from 800 religious orders. It was conducted at Rush University Medical Center. The research examined the relationship between various factors (including lifestyle, environment, and genetic predispositions) and the development of Alzheimer's disease and other age-related cognitive decline factors among Catholic clergy members. One of the findings of the Religious Order Study demonstrated that participation in religious activities, such as prayer, meditation, and religious services, was associated with a reduced risk of developing Alzheimer's disease and other forms of dementia. Additionally, the research also found that those who engaged in more cognitive activities, such as reading, writing, and solving puzzles, exhibited a lower risk of cognitive decline. Overall, the Religious Order Study revealed the potential protective effects of religious and cognitive activities against Alzheimer's disease and age-related cognitive decline (Bennett et al. 2018). Notable research has been derived from both the Religious Order Study and the Nun Study.

As the global population ages, with one in five individuals projected to be older adults by 2050, understanding the factors influencing successful aging becomes more crucial. Research suggests that Women Religious, clergy, and deeply religious individuals may age more successfully and live longer, healthier lives. This article review explores the intersection of spirituality, religious practices, and successful aging, addressing the increasing demographic of older individuals. By 2050, the elderly population is projected to reach nearly 2.1 billion, prompting a need to understand the role of spirituality and religion in the aging process (World Health Organization 2019). Religious and spiritual communities are recognized for their lifestyle practices, which often provide crucial emotional and mental support. Numerous studies emphasize the association between religious or spiritual practices and better physical health, increased longevity, improved mental health, and enhanced social support (Valino 2021). These findings underscore the holistic benefits of incorporating spirituality into the lives of older adults. A unique perspective has been presented by Schott and Krull (2019) through a small cross-sectional study involving nuns, monks, master athletes, and sedentary controls. Nuns and monks demonstrated superior cognitive performance compared to master athletes, and fitness levels were associated with inhibitory control and physical activity with demanding working memory tasks. This unique study provides novel evidence, shifting the focus from age-related cognitive losses to the study of physically and cognitively engaged individuals. The lifestyle stability observed in nuns and monks is correlated with better cognitive performance, emphasizing the importance of adhering to various lifestyle factors in later life (Clare et al. 2017). The communal and homogeneous living of nuns, Catholic sisters, and monks may promote healthy

behaviors such as low alcohol and tobacco consumption, daily prayer, and meditation, thereby potentially increasing brain, and cognitive reserve capacity.

Current evidence suggests that religious and spiritual practices may foster positive psychological emotions and virtues, such as hope, optimism, forgiveness, and gratefulness, which may serve as mediators for aging well (Sutin et al. 2021). Religious communities offer opportunities for social engagement and community support, and members often act as caregivers when needed throughout the life cycle. Preliminary research indicates that religiosity and spirituality may also mitigate stress and potentially reduce inflammatory markers (Chen et al. 2018). Studies on improved cognition in older adults have found similar connections to well-being. Higher engagement in religious and spiritual practices is linked to slower cognitive and behavioral decline and better overall functioning in older adults living with dementia (Britt et al. 2022). One study employing a spiritual intervention involving reminiscence demonstrated improved cognitive function among participants with dementia (Wu and Koo 2016).

Spirituality continues to emerge as a significant area of inquiry in aging research (Koenig 2013; Atchley 2009). According to Bailey et al. (2016), spirituality has been broadly defined as dynamic connectedness to oneself, others, or the divine in constructs of meaning. Weathers et al. (2016), in a similar manner, describe spirituality as connectedness and transcendence beyond self, everyday living practices, and suffering. Spirituality, religiosity, and longevity are concepts that have been investigated in numerous studies in association with the process of aging. Spirituality and religion often appear interchangeably in research (Yeşilçınar et al. 2018). An increasing number of studies have examined the complexity and interdisciplinary connection between spirituality, religiosity, health, and quality of life. According to Joseph et al. (2017), spirituality should be understood as a more general, unstructured, and personalized phenomenon that occurs naturally. The individual may also seek a connection to a higher power. Spirituality and religion may further help shape perceptions of health as well as interactions with others (Bożek et al. 2020).

Defining spirituality as a standalone concept can be challenging. However, the recent nursing literature conceptualizes spirituality as the quest to find purpose in life, foster connectedness, seek transcendence, and immerse in realms deeper than religion alone (Timmins and Caldeira 2017). This concept of spirituality may extend beyond religion and may provide non-believers with purpose, meaning, and reverence (Cockell and McSherry 2012). According to Cockell and McSherry (2012), spirituality gives individuals a sense of congruence and helps them seek answers to many of life's infinite questions. This may provide solace during times of stress, illness, or end-of-life situations. Similarly, O'Brien et al. (2019) emphasize that spirituality is a personal concept involving an individual's notions of transcendence, the divine, and mind–body–spirit connection. Malone and Dadswell (2018) discovered that religion and spirituality may provide strength, comfort, hope, and a sense of community, suggesting that spirituality and successful aging influence older individuals' health and well-being. Spirituality is also recognized as a crucial aspect of nursing care that is supported hypothetically and validated in practice (Veloza-Gómez et al. 2017). Spirituality may strengthen psychological well-being and confirm that existential thinking and life meaning may be related to overall beneficial mental health outcomes (Giannone and Kaplin 2020). Life meaning among older adults often plays a pivotal role in physical and mental health, thus impacting the aging experience (Greenblatt-Kimron et al. 2022).

Researchers have acknowledged that spiritually grounded perspectives are related to better tolerance to emotional and physical stress, successful aging, and the ability to cope with serious diseases and isolation (Le et al. 2019; Sharma et al. 2019). While genetics plays a role in the aging process, the extent to which lifestyle practices can mitigate genetic factors remains a key focus for researchers. Conditions like Alzheimer's, cancer, heart disease, and diabetes, often associated with advancing age, prompt the promotion of healthy lifestyle behaviors to improve overall health in the aging population. Another study conducted by Can Oz et al. (2022) identified the meaning and perception regarding religion and spirituality in older age. Nineteen participants, ranging in age from 65 to 88 years old, were

included in the study. Of these, five were male and fourteen were female. Semi-structured interviews were utilized, garnering four main themes. The themes emerged as the meaning of spirituality, spiritual practices, the effect of spirituality, and the meaning of growing older. The results indicate that the participants viewed entrance into older adulthood as a process that enhances interactions with others and intensifies compassion. Overall, the results indicate that spirituality may play a crucial role in the lives of older adults, giving them a deeper ability to cope with and clarify life meaning. The inclusion of spirituality in the lives of older adults can provide support and facilitate positive emotions.

The concept of aging remains multifaceted. Currently, many researchers view aging as socially constructed and culturally defined. Religiosity may offer numerous advantages to believers and associated with improved coping skills, enhancements in quality of life, finding meaning in life, and sustaining hope during challenging circumstances or experiencing life in older age (Dolcos et al. 2021). Qualitative evidence suggests the utilization of religious coping strategies during difficult life experiences may prove to be beneficial. Research also underscores the efficacy of cognitive reappraisal in mitigating the impact of distressing emotions on overall well-being. Numerous studies have correlated beliefs in the divine with positive self-image, outlook, and improved mental well-being and satisfaction with life (Shafranske 2023). According to Hood et al. (2018), religiosity contributes to a sense of meaning in life. The research of Shiah et al. (2015) found that religious people have a higher awareness of coherence and purpose in life than non-religious people. A study of 33 midwestern Catholic sisters residing in a community was described, where prayer sessions practiced by the sisters were monitored using brain electrodes (Barcelona et al. 2020). The overall results suggest that prayerful meditation increased positive left frontal and central brain activity as the sisters aged. Future research may be essential in capturing the mediating role of age and the connection to meditation, prayer, and cognitive outcomes.

The concept of successful aging is now shifting from a strictly bioscientific definition to a more holistic perspective. This shift embraces subjective aspects of the aging process (Jyväkorpi et al. 2019). While cognitive, social, emotional, and personality factors have been identified as predictors of longevity, the role of motivation has been largely overlooked. Intelligence, conscientiousness, openness, extraversion, positive emotions, and social support have all been linked to increased life expectancy (Gottfredson and Deary 2004; Fry and Debats 2009; Danner et al. 2001; Brown and Ryan 2003). According to Dennis and Thompson (2014), success in aging is not influenced by the genetic predisposition of an individual but rather by factors such as physical activity levels, social interactions, and attitude towards life in general. Moreover, studies indicate healthy lifestyles, including moderation in eating, proper hydration, regular exercise, purposeful living, spirituality, and maintaining social connections, play a crucial role in mitigating age-related limitations. Spirituality has been found to positively impact the lives of older adults, enhancing mental health and overall well-being. It provides a foundation for finding meaning in life, coping with adversity and illness, and navigating the physical changes that come with aging (Malone and Dadswell 2018).

From a sociological perspective, specific social groups who are dedicated to religious life engage with spirituality and religiosity as a resource to cope with chronic diseases, contributing to a higher quality of life and well-being (Iannello et al. 2022). Religiosity has been linked to self-enhancement tendencies and socially desirable responses (Sedikides and Gebauer 2021). Religion and spirituality have been defined in largely ambiguous terms, and in research, they are generally assessed through single-item self-reported measures, such as church attendance, that lack construct validity (Flannelly 2017). According to Flannelly (2017), belief in an afterlife is common among religious individuals. Afterlife beliefs have been found to be positively correlated with life satisfaction (Cohen et al. 2006). Various studies have consistently demonstrated the positive association between spirituality and mental health, supporting the improvement in life quality, social interactions, and positive attitudes toward aging (Abdel-Khalek et al. 2019). Belief in the transcendent or divine continues to emerge as a significant dimension contributing to the meaning of life,

resilience, and health promotion among older adults (Abdel-Khalek et al. 2019). Aging as a multilayered process encompasses multiple levels of change on a biological, psychological, communal, and economic level. Common challenges frequently stem from social isolation, chronic disease, financial difficulties, and limitations in both physical and mental health well-being, which are more commonly experienced during the aging process. Providing older adults with adequate resources designed to support the aging process may offset any adverse physical and mental health outcomes and thereby support healthier aging. Research overall suggests religiosity is frequently linked with enhanced mental well-being and a sense of community.

Recent research in the field of gerontology accentuates practices that may help older adults achieve successful aging, promoting positive functioning and happiness in later life stages. Successful aging and well-being across adulthood have become central themes in research, with studies highlighting their connections to various positive life outcomes and decreased mortality rates. Maintaining well-being throughout older age is essential, and various studies have demonstrated that subjective and spiritual well-being are affected by a multitude of factors. These factors include indicators of social and stable economic status, education, marital status, social support, and religiosity (Pronk et al. 2021). Malone and Dadswell (2018) suggest that positive aging, or, alternatively, successful aging, includes various circumstances that older adults may use to confront challenges that are associated with aging. Confronting these challenges enables them to age in a more positive manner. Malone and Dadswell (2018) utilized a qualitative focus group design of 14 older adults. The research explored the role and importance of religious practice, spirituality, and/or beliefs practiced in their everyday lives and their connection to positive aging. The results provided evidence that religion, spirituality, and/or beliefs were found to play several roles in the everyday lives of older adults, namely, providing a sense of strength, a sense of community, comfort, and hope in difficult times with the premise of facilitating the aging process.

Studies examining the brain lifestyles of Buddhist monks provide further evidence suggesting that daily meditation might offer protection against age-related brain changes. Research investigating the effects of meditation continues to gain attention. Studies to date have provided evidence that meditative practices may slow the attrition of brain deterioration stemming from a loss of the brain's gray matter (Tang et al. 2020). In Tang et al.'s longitudinal study, conducted using a randomized design, it was observed that the loss of the brain's gray matter occurs naturally with aging. To date, research investigating the link between mindfulness meditation and age-related gray matter decline has predominantly been cross-sectional. For example, studies have noted that the inverse relationship between age and gray matter volume is less pronounced in long-term meditators compared to non-meditators in the wider population (Luders et al. 2016). According to Smith et al. (2019), meditative practices may have the potential to preserve cognitive health among older adults. Additionally, it has been observed that the brain age of long-term meditators is lower than that of non-meditators of the same chronological age (Luders et al. 2016). The results prompt a more systematic longitudinal evaluation of the relationship between extensive meditation practice and brain aging.

Anthropologist Anna Corwin's ethnographical research on aging nuns challenges common assumptions about successful aging. Despite being held up as models of successful aging, the nuns themselves view aging as a natural process. Their lifestyle, characterized by communal living, prayer, and caregiving for each other, contributes to their longevity, physiological health, and happiness. The provided literature also acknowledges the need for continued longitudinal studies to observe changes in the quality of life and aging over the years in religious communities. According to Corwin (2021), the culture in the convent emphasizes the importance of daily prayer as a connection to the divine. This is also a way to communicate inner needs, feel mutual support, and be embraced by God. Prayer and meditative practice, including contemplation, are viewed as positive practices that promote mental and physical health. Finally, prayer allows the nuns to meet death in an equally

positive way. Dr. Corwin's research looks at a vast range of factors that may improve aging. For example, consistent nutrition, higher education, positive emotions associated with longevity, and a positive view of the broader world.

Experimental studies on human mortality conducted at the Institute for Demography of the Austrian Academy of Sciences revealed that religious individuals tend to live approximately five years longer, on average, than non-believers (Luy 2021). The primary objective of the study was to investigate the fundamental determinants of health and longevity. Primary investigator Mark Luy suggests that adhering to a regular daily routine, maintaining a balanced diet, and engaging in prayer may positively influence the health of individuals in religious communities. To explore the secrets of successful aging, researchers examined life expectancy among monks residing in monasteries across Austria and Germany. The initial phase of the survey involved 1158 participants, including 622 women and 536 men, representing 16 different religious orders in the two countries. Surprisingly, the study found that religious women lived only one year longer than their male counterparts. This conclusion contradicted the normative trend that women typically outlive men by a significant margin. This result suggests that, regardless of gender, by adopting monastic lifestyles, including adhering to routines, dietary practices, and prayer, the disparity in average life expectancy between men and women would diminish to just one year.

Amiotte-Suchet and Anchisi (2024) utilized an ethnographical approach and recruited subjects from 16 monastic communities. The research was an immersion into monastic life in both eastern France and Switzerland. Furthermore, monks and nuns choose to live within their traditions of a successful, spiritual, and communal life and hope to carry forward this model (Amiotte-Suchet and Anchisi 2024). In summary, the view of handling their aging brothers or sisters may also be seen as beneficial into old age. Amiotte-Suchet and Anchisi (2024) utilized extracts from field journals of those in the study, which demonstrated monks and nuns manage their lives with pragmatic solutions, prayer, community, tranquility, and maintaining spiritual virtuosity. According to Schott and Krull (2019), stability of lifestyle as evidenced among religious communities may also confer benefits to maintaining cognition in older adults. Further longitudinal studies are recommended to examine the causal relationship between lifestyle stability and cognitive function.

Furthermore, a belief in the transcendent or the divine may influence the quality of life of older adults, contributing to resilience and positively impacting health promotion (Manning et al. 2019). Studies regarding religion and spirituality, including positive psychology, demonstrate a positive association between religiosity, spirituality, and subjective well-being, life meaning, optimism, and positive self-esteem (Abdel-Khalek et al. 2019). Considerable research has demonstrated that the elderly are more religious and that religiosity is associated with better health and lower mortality (Lechler and Sunde 2020). Evidence suggests prayer and religiosity appear to have a protective effect on older adults' cognitive function (Hill et al. 2006). Studies on meditation and prayer have also shown that both could have a positive effect on the cognitive function of older adults (Gard et al. 2014). Research on many platforms continues to demonstrate a connection between religiosity, spirituality, and subsequent self-enhancement that may facilitate individuals to respond in a socially desirable manner (Sedikides and Gebauer 2021). A significant number of studies continue to show an association exists between cognitive performance and lifestyle behaviors such as spirituality in older adults (Clare et al. 2017). Nuns and monks represent a religious community of relatively homogenous individuals, and their lifestyle practices, such as daily prayer and meditation, might lead to increases in cognitive reserve capacity (Bowen and Luy 2018). As the aging population continues to grow, more research is needed that can identify and describe the complex interaction of meaningful genetic, biological, and lifestyle practices, especially among certain communities (Dixon and Lachman 2019).

**Conclusions**

Based on the provided literature review, the conclusion can be drawn that spirituality, religious practices, and communal living have a significant impact on successful aging and overall well-being among older individuals, particularly exemplified by Catholic nuns,

or Women Religious, and monks. The association between religion, prayer, consistent lifestyle practices, and well-being seems to be robust among religious communities. The lifestyle factors associated with religious communities, such as daily religious practice, communal living, robust social support, and spirituality, contribute to positive physical and psychological health outcomes, including increased longevity and enhanced mental health. Moreover, studies consistently demonstrate the positive correlation between spirituality, religiosity, and mental health, supporting the improvement in life quality, social interactions, and positive attitudes towards aging. The belief in the transcendent or divine emerges as a significant dimension contributing to the meaning of life, resilience, and health promotion among older adults.

This review underscores the necessity for additional research to comprehensively grasp the concepts of religiosity and spirituality for successful aging. Aging research that focuses on these concepts may highlight the positive impacts on the health and elevation in the quality of life of older individuals. Enhancing our comprehension of successful aging may also propel future research in the realms of public health, nursing, and gerontology, thereby supporting the advancement of preventative and therapeutic approaches for older adults. This review also emphasizes the need for continued longitudinal studies to observe changes in the quality of life and sense of life over the years among older adults engaged in religious and spiritual practices. Understanding the impact of spirituality and religious practices on the aging process becomes increasingly essential as the global older adult population continues to grow. In summary, the conclusion drawn from the literature suggests that spirituality and religiosity play a crucial role in successful aging, promoting overall physical health, mental well-being, resilience, and a positive outlook on life among older adults.

**Funding:** This research received no external funding.

**Data Availability Statement:** No new data were created or analyzed in this study. Data sharing is not applicable to this article.

**Conflicts of Interest:** The author declares no conflict of interest.

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
