# Peer review of "The Intersection of Spirituality, Religiosity, and Lifestyle Practices in Religious Communities to Successful Aging: A Review Article"

_religions, doi:10.3390/rel15040478_

Round 1

Reviewer 1 Report

Comments and Suggestions for Authors

The author provides a strong study for the topic, an interesting and timely research. Many statements jump off the page to me.

For example, "Research indicates that Catholic nuns experience extended lives and enhanced well-being, potentially attributed to lifestyle factors promoting successful aging."

The author looks at current research and concludes, "Current evidence suggests that religious and spiritual practice may foster positive psychological emotions and virtues, such as hope, optimism, forgiveness, and grateful ness, which may serve as mediators to aging well "A significant shift is noted, " The concept of successful aging is now shifting from strictly bioscientific definition to a more holistic perspective."

The References provides a seemingly exhaustive and comprehensive list of relevant material.

The author follows a proper approach to the research.

And finally the author states, "In summary, the conclusion drawn from the literature suggest spirituality and religiosity play a crucial role in successful aging, promoting overall physical health, mental well-being, resilience, and a positive outlook on life among older adults."

Author Response

Dear Reviewer,

I would like to express my sincere gratitude for taking the time to read my manuscript. Your feedback is immensely valued and appreciated. My research of late and specifically this manuscript was meant to capture the literature for the purposes of a review. The aim was to provide a comprehensive overview of existing research on the topic as opposed to a hypothesis or method. When a study was included, the available method was stated. Within the manuscript, I was able to include the number of participants, methodology of the research (and for ex., longitudinal, ethnographical research, qualitative vs. quantitative, structure of the research design)

Thank you again.

Respectfully,

Author

Reviewer 2 Report

Comments and Suggestions for Authors

This work is a helpful contribution to the overall theme of "Spirituality and the aging process"

The manuscript is a well written Review of existing works.

Consider writing a paragraph that states your hypothesis and describes your chosen method/approach if possible  

Author Response

(The authors gave the same response as above.)

Reviewer 3 Report

Comments and Suggestions for Authors

This article is very important and interesting. The research is based on religious communities of men and women in the Catholic tradition. The author might include as areas for further research societal communities where religion and spirituality are critical elements; e.g. Amish or Mennonite communities. Further, and probably more difficult, it would be interesting to research individuals for whom spirituality is important and who have found informal communities of support. As formal religious affiliation declines, there seems to be almost an explosion of informal, independent spiritual communities. How do they impact longevity and 'happiness'?

Author Response

Dear Reviewer,

I would like to express my sincere gratitude for taking the time to read my manuscript. Your feedback was immensely appreciated.

Kind regards,

Author